# Comprehensive analysis of immune-related genes reveals diagnostic biomarkers and molecular subtypes in diabetic retinopathy

Lin Mu[1,2,3⊕], Zhe Wang[4⊕], Yaojiani Cao[1,2,3], Xuejun Wang[1,2,3*], Xingtao Zhou[1,2,3*]

1 Eye and ENT Hospital of Fudan University, Shanghai, China, 2 Shanghai Research Center of Ophthalmology and Optometry, Shanghai, China, 3 Institute of integrative opthalmology and otolaryngology, Eye and ENT Hospital of Fudan University, Shanghai, China, 4 The First Affiliated Hospital of Naval Medical University, Shanghai, China

⊕ These authors contributed equally to this work.
* wangxuejun9527@126.com (XW); doctzhouxingtao@163.com (XZ)

## Abstract

Diabetic retinopathy (DR) is a common microvascular complication of diabetes and one of the primary causes of vision loss; however, its pathogenic mechanisms remain largely unresolved. In recent years, immune dysregulation has been increasingly recognized to be closely associated with DR progression. In the present study, we integrated two GEO datasets to identify immune-related differentially expressed genes (DEGs) associated with DR. After batch effect correction and differential expression analysis, a total of 123 immune-related DEGs were identified. Functional enrichment analysis demonstrated that these genes are primarily associated with immune activation pathways, such as T-cell receptor signaling, natural killer cell–mediated cytotoxicity, and IL-17 signaling. Using least absolute shrinkage and selection operator (LASSO) regression, five key genes (IL10RA, PLAUR, PLAU, VTN, and VGF) were identified and used to construct a diagnostic model with excellent predictive performance. Protein–protein interaction and immune infiltration analyses revealed that these genes are closely associated with immune cell activity, particularly the increased infiltration of resting CD4 memory T cells and M2 macrophages in the DR retina. To validate these findings, an STZ-induced diabetic mouse model was used, with key genes further validated at the protein level via Western blot. In addition, consensus clustering was used to stratify patients with DR into two molecular subtypes with distinct immune characteristics and pathway enrichment patterns. Overall, our study elucidates the immune-related mechanisms underlying DR and highlights PLAUR, PLAU, and VGF as potential immune-associated biomarkers, providing a theoretical basis for precision diagnosis and development of targeted immunotherapeutic strategies for DR.

**Data availability statement:** The main data used in this study are publicly available from the Gene Expression Omnibus (GEO) database under accession numbers GSE60436 and GSE102485.

**Funding:** 2025 Shanghai Integrated Traditional Chinese and Western Medicine Collaborative Guidance Program for General Hospitals (Grant No. ZXXT-202512), 2025 National Major Difficult and Critical Diseases Integrated Traditional Chinese and Western Medicine Clinical Collaboration Project (Optic Atrophy No. 13), and National Major Difficult and Critical Diseases Integrated Traditional Chinese and Western Medicine Clinical Collaboration Project (ZDYN-2024-A-052). The funders had no role in study design, data collection and analysis, decision to publish, or preparation of the manuscript.

**Competing interests:** The authors declare no competing interests.

**Abbreviations:** CDF, cumulative distribution function; DEGs, differentially expressed genes; DR, diabetic retinopathy; GEO, gene expression omnibus; GSEA, gene set enrichment analysis; IL10RA, interleukin-10 receptor subunit alpha; KEGG, kyoto encyclopedia of genes and genomes; LASSO, least absolute shrinkage and selection operator; NPDR, non-proliferative diabetic retinopathy; PDR, proliferative diabetic retinopathy; PLAUR, plasminogen activator, Urokinase receptor; PLAU, plasminogen activator, Urokinase; VTN, vitronectin; VGF, nerve growth factor inducible.

# 1 Introduction

Diabetic retinopathy (DR) is one of the major eye diseases that cause blindness in adults worldwide [1]. By 2045, 160 million people worldwide will have DR, resulting in the significant consumption of global medical resources [2]. Although various treatments, including drug injection, laser therapy, and surgery, have improved DR symptoms to some extent, these approaches are often accompanied by notable side effects [3]. Thus, elucidating the molecular mechanisms underlying DR pathology is essential for developing novel therapeutic strategies [4].

Emerging evidence suggests that immune dysfunction plays a key role in DR pathogenesis [5]. Therefore, a comprehensive understanding of immune cell dysregulation at the molecular level may contribute to the development of improved therapeutic approaches [6].Numerous studies have investigated the molecular mechanisms of DR and have revealed that the immune response is closely involved in the pathogenesis of this disease [7]. For example, DR is thought to be driven by low-level, continuous leukocyte stimulation, which may lead to capillary occlusion and repeated vascular spasms, ultimately contributing to retinal vascular alterations [8]. In addition, hyperglycemia has been shown to increase CD40 levels in monocytes and platelets, potentially promoting leukocyte accumulation and ICAM-1 overexpression, which may contribute to capillary inflammation [9]. Under high-glucose conditions, retinal pigment epithelial cells upregulate CD95 ligand expression, which may interact with CD95 receptors on retinal microglia and induce apoptosis [10]. In vivo studies have shown that IL17A can regulate retinal Müller cells and may promote apoptosis of retinal ganglion cells in DR mice [11]. Immune-related targets and cells are involved in the pathogenesis of DR; thus, regulating the immune response of cells or cytokines may become a new target for treating DR [12].

Despite these advances, the comprehensive landscape of immune-related gene expression patterns and molecular subtypes in DR remains poorly characterized. To address this gap, we analyzed DR patient data from the GEO database and identified immune-related differentially expressed genes (DEGs) using GO and GSEA. Key biomarkers and gene interactions were identified through Lasso-logistic and Spearman analyses, while immune cell infiltration was evaluated using CIBERSORT. Consensus clustering was used to define immune-related molecular subtypes, and ssGSEA to explore differences in biological processes between subgroups. Finally, we validated the expression of hub genes in an STZ-induced diabetic mouse model.

Although previous studies have reported associations between individual immune-related genes and DR, a comprehensive analysis integrating multiple datasets to identify immune-related molecular subtypes and diagnostic biomarkers remains limited. In this study, we systematically analyzed two GEO datasets to identify immune-related DEGs, constructed a diagnostic model using LASSO regression, and validated key genes in an STZ-induced mouse model. Our integrated approach provides a more complete picture of immune dysregulation in DR and identifies potential subtype-specific biomarkers and therapeutic targets.

## 2 Materials and methods

### 2.1 DR data acquisition and preprocessing

The retina-related gene expression profiling data and corresponding clinical information of diabetic patients with data numbers GSE60436 and GSE102485 were downloaded from the GEO database (https://www.ncbi.nlm.nih.gov/geo/) [13,14]. The sample source was *Homo sapiens*, and the two dataset sequencing platforms were GPL6884 and GPL18573. GSE60436 contained 9 samples—3 healthy and 6 DR samples. The GSE102485 dataset included 30 samples—16 healthy and 14 DR samples.

The inclusion criteria for dataset selection were as follows: (1) retinal tissue samples from Homo sapiens; (2) clearly defined DR and normal control groups; (3) availability of complete expression data; and (4) sample size ≥ 3 per group.

The "cbind" function in R was used to integrate the expression matrices of the two datasets. Batch effects between the datasets were corrected using the R package sva, and the data were standardized using log2(X + 1).

### 2.2 Identification of DEGs related to DR

To analyze the effect of the gene expression level of immune-related genes on DR, the R package limma was used to screen the DEGs between disease samples and normal samples in the integrated dataset [15]. An absolute value of log2fold change (log2FC) > 1 and $p < 0.05$ were set as the thresholds for differential genes. Multiple testing correction was performed using the Benjamini–Hochberg method, and an adjusted false discovery rate (FDR) < 0.05 was considered statistically significant. The differential gene expression results were visualized using a volcano map and heatmap.

### 2.3 Comparison of biological characteristics between DR and normal retinas

Gene Ontology (GO) analysis is a common method for large-scale functional enrichment studies, including biological processes (BP), molecular functions (MF), and cellular components (CC). The clusterProfiler package of R was used to analyze the DEGs with GO annotation, and a critical value of FDR < 0.05 was considered to indicate statistical significance [16]. Gene set enrichment analysis (GSEA) was used to identify pathways that were consistent between the DR group and normal retinal tissue and was performed using the R package clusterProfiler. $p < 0.05$ was considered to indicate statistical significance.

### 2.4 Construction of the forest model and nomogram model

To accurately screen biomarkers related to DR, we used 1000 iterations of the least absolute shrinkage and selection operator (LASSO) model for dimensionality reduction screening. The objective function of the Lasso logistic regression model is as follows:

$$min \int (\alpha_0, \alpha | X_i, Y_i + \lambda ||\alpha||_1).$$

(1)

$\lambda$ represents the penalty coefficient in the function, and the optimal $\lambda$ can be selected by 10-fold cross-validation. $||\alpha||_1$ is defined as the sum of the absolute values of each vector element. R packet glmnet is used for Lasso logistic regression [17]. The risk score formula is established through the normalized gene expression values weighted by the penalty coefficient of characteristic genes:

$$riskScore = \sum_i Coefficient\ (gene_i) * mRNA\ Expression\ (gene_i).$$

(2)

Given that immune-related pathways play an essential role in DR, the biological functions of immune-related genes may differ between samples from healthy individuals and those with DR; thus, it is feasible to construct a diagnostic model of immune-related differential genes. The forest model was used to predict the occurrence of DR by selecting candidate DEGs from among the differentially expressed immune-related genes. The candidate differentially expressed immune-related genes were included in the model, and the LASSO algorithm was used to analyze the dimensionality reduction and obtain immune-related characteristic genes. The expression values of each normalized gene weighted by the penalty coefficient of characteristic genes are weighted, and a risk scoring formula is established.

$$\text{riskScore} = \sum_i \text{Coefficient } (\text{gene}_i) * \text{mRNA Expression } (\text{gene}_i).$$

(3)

Finally, a line chart model is constructed based on the selected differentially expressed immune-related genes to predict the prevalence of diabetes.

## 2.5 Friends analysis

Friends analysis is a method used to explore the interaction between genes and to screen essential functional genes further. In this study, friends analysis was conducted using the R packet GOSemSim [18]. Through analysis of the LASSO algorithm, the essential functional genes with the strongest correlation between differentially expressed immune-related genes and other characteristic genes were screened as possible hub genes.

## 2.6 Spearman analysis of hub genes

To explore the correlation between hub genes, the R package "cowplot" was used to perform a Spearman analysis of crucial genes, and heatmaps, scatter plots, and correlation curves were constructed. $p < 0.05$ was considered to indicate statistical significance. R packet RCircos was used to map six genes on the chromosomes [19]. This R packet provided the chromosome data. The chromosomal location information of genes was downloaded and obtained from the ENSEMBL database [20].

## 2.7 Construction of protein-protein interaction network (PPI)

The STRING database (https://cn.string-db.org/) was used to construct a PPI network based on the screened differential genes of DR. Cytoscape (v3.7.2) was used to visualize the PPI network model. The clustering coefficient network clustering algorithm explores protein complexes or functional modules from complex protein networks [21]. The clustering coefficient algorithm was used to mine the gene with the highest score in the PPI network as the core gene of DR.

## 2.8 CIBERSORT analysis

CIBERSORT (https://cibersort.stanford.edu/) [22] can be used to evaluate the infiltration of immune cells in sequenced samples based on the gene expression feature set of 22 known immune cell subtypes. In this study, immune cell infiltration status in an integrated dataset of DR patient retinas was evaluated using the CIBERSORT algorithm. The differences in immune cell infiltration between DR patient samples and normal samples were then examined using the Wilcoxon test, with the significance level set at $p < 0.05$.

## 2.9 Molecular subtypes of DR

Consistent clustering is an algorithm based on resampling used to identify each member and its subgroup number and verify the rationality of clustering. R packet ConsensusClusterPlus was used to identify different immune-related molecular patterns based on the crucial differentially expressed immune-related genes in DR samples and healthy samples [23].

## 2.10 Identification of DEGs among different modes of DR

To analyze the impact of different immune patterns on DR, the R package limma was used to perform differential gene analysis on the immune pattern subgroups in the integrated dataset. Significant DEGs were screened based on thresholds of log2fold change (log2FC) absolute value > 1 and $p < 0.05$. Genes with log2FC > 1 and $p < 0.05$ were considered upregulated differentially expressed genes, whereas those with log2FC < −1 and $p < 0.05$ were considered downregulated differentially expressed genes. The results of the DEGs are presented using a volcano plot.

## 2.11 Evaluation of biological characteristics among patients with different DR models

To investigate differences in biological processes among different groups, we utilized a gene expression profile dataset of patients with DR. We performed gene set enrichment analysis using single sample gene set enrichment analysis (ssGSEA) [24]. SsGSEA is a computational method that evaluates significant differences between two biological states in a particular gene set. This approach is often employed to estimate changes in pathway and biological process activity in expression datasets, with $p < 0.05$ considered to indicate statistical significance. Initially, we labeled various types of infiltrating immune cells, including activated CD8 T cells, activated dendritic cells, gamma delta T cells, natural killer cells, regulatory T cells, and other human immune cell subtypes. Next, we used the enrichment score calculated by ssGSEA to quantify the relative abundance of immune cell infiltration in each sample. Finally, we used the ggplot2 package to visualize the distribution of immune cell infiltration in different risk groups and disease subtypes of DR. Additionally, we employed the R package pheatmap to generate correlation heatmaps to illustrate the relationship between mitochondrial autophagy-associated LASSO genes and immune cells.

## 2.12 Construction of a diabetic mouse model

C57BL/6 male mice (6 weeks old, weighing 20–25 g) were purchased from Beijing Vital River Laboratory Animal Technology Co., Ltd. (Beijing, China). The animals were housed in a barrier environment at a temperature of 25 °C, with a dark photoperiod of 12:12 h and free access to food and water. The Ethics Committee of Longhua Hospital Shanghai University of Traditional Chinese Medicine approved all the animal experiments (approval No. P 2022026).

A total of 20 mice were randomly divided into a normal group and a DR group.Sample size (n = 10 per group) was determined based on previous similar studies and the principles of the 3Rs (Replacement, Reduction, Refinement) to minimize animal usage while ensuring statistical power.

After one week of acclimatization, the DR group received an intraperitoneal injection of STZ (50 mg/kg, prepared in 100 mM citrate buffer) for five days [25]. One week after injection, the fasting blood glucose levels of the mice were evaluated on three consecutive days using tail vein blood and a glucometer. When the blood glucose concentration was greater than 300 mg/dL, a diabetic model was established [26].

Body weight and blood glucose levels were monitored weekly during the experimental period. Retinal tissues were collected 2 months after confirmation of diabetes. Mice were anesthetized with intraperitoneal sodium pentobarbital (50 mg/kg) prior to tissue collection. Euthanasia was performed by cervical dislocation under deep anesthesia. All efforts were made to minimize animal suffering.

## 2.13 Western blot analysis

Retinal tissues from all mice at the 2-month time point after diabetes confirmation (normal group, n = 10; DR group, n = 10) were collected for protein extraction. Lysis buffer was used to separate total proteins from the mouse retina. After centrifugation at 12000 × g for 15 min, the protein concentration in the supernatant was detected using a BCA protein assay kit (Biyuntian, China). Total protein (20 µg) was separated by electrophoresis on a 10% SDS-polyacrylamide gel and transferred to a PVDF membrane (Millipore, Billerica, MA, USA). The membrane was subsequently immersed in 5% skim

milk for 1 h at room temperature. The membrane was separately incubated with anti-uPA (1:1000, Catalog # 15800, Cell Signaling Technology), anti-uPAR (1:1000, Catalog # 12863, Cell Signaling Technology), and anti-VGF (1:500, Catalog # PA5–20523, Thermo Fisher) at 4 °C for 12 hours. The membrane was subsequently incubated with an HRP-linked antibody (1VR 1000; Catalog # 7074; Cell Signaling) for 2 hours. After exposure to a chemiluminescence substrate (Bio-Rad, Hercules, CA, USA), the immunoreactive signal was analyzed by a gel imaging system (Bio-Rad, CHEMI DOC MP, USA) and quantified by ImageJ.

### 2.14 Statistical analysis

All data were processed and analyzed (version 4.1.1) in R software. For the comparison of two groups of continuous variables, the statistical significance of standard distribution variables was estimated by the independent Student's t test, after assessment of data normality using the Shapiro–Wilk test and homogeneity of variance using Levene's test.The difference between nonnormally distributed variables was analyzed using the Mann–Whitney U test (Wilcoxon rank-sum test). The chi-square test or Fisher's exact test was used to compare and analyze the statistical significance between the two groups of classified variables. The correlation coefficients between different genes were calculated by Pearson correlation analysis, under the assumption of a linear relationship between variables. All the P values were bilateral, and $p < 0.05$ was considered to indicate statistical significance.No missing data were identified in the analyzed GEO datasets; therefore, no additional procedures for handling missing data were required.

## 3 Results

### 3.1 Differential expression of genes in the diabetic retina and normal retina

First, we processed the DR datasets GSE60436 and GSE102485 using the R package sva to remove the batch effect. This resulted in the creation of combined datasets. To assess the effectiveness of batch effect removal, we generated a principal component analysis (PCA) plot to compare the datasets before and after batch removal (Fig 1). The results of the PCA demonstrated that the batch effect in the DR dataset was eliminated mainly after the batch removal process, as evidenced by the improved separation of samples in the plot. A total of 1055 DEGs were obtained by differential analysis between DR samples and healthy samples, including 560 upregulated DEGs and 495 downregulated DEGs (Fig 2A, C).

To explore the role of immune-related genes in DR, 123 immune-related differentially expressed genes were obtained from the intersection of immune-related genes and differential genes (Fig 2B). To analyze the functional differences between DR and control samples, we analyzed the effects of DEGs on the biological functions of patients. GO functional annotation of the DEGs (Fig 2D–F) revealed that these DEGs were mainly enriched in the biological processes of cell chemotaxis and T-cell activation, the cellular component of the plasma membrane, and the molecular function of receptor ligand activity.

### 3.2 GSVA of diabetic retina and normal retina

The results of GSVA showed that cytokine interaction, T-cell receptor signaling pathway, NK cell-mediated cytotoxicity, leukocyte transendothelial migration, the Toll-like receptor-related pathway, FC γ-mediated phagocytosis, the IL-17 signaling pathway, and extracellular neutrophil trap formation were significantly enriched in patients with DR, suggesting a possible inflammatory response at the lesion site (Fig 3).

### 3.3 Construction of the prediction model

The differential genes related to immunity were screened by 1000 Logistic Lasso analysis, and the optimal lambda value of the model was used as the screening condition. The results showed that 5 characteristic genes were most closely related to DR: PLAUR, PLAU, VTN, IL10RA, and VGF, which appeared 795 times after 1000 cycles (Fig 4A–B). A prediction model

 

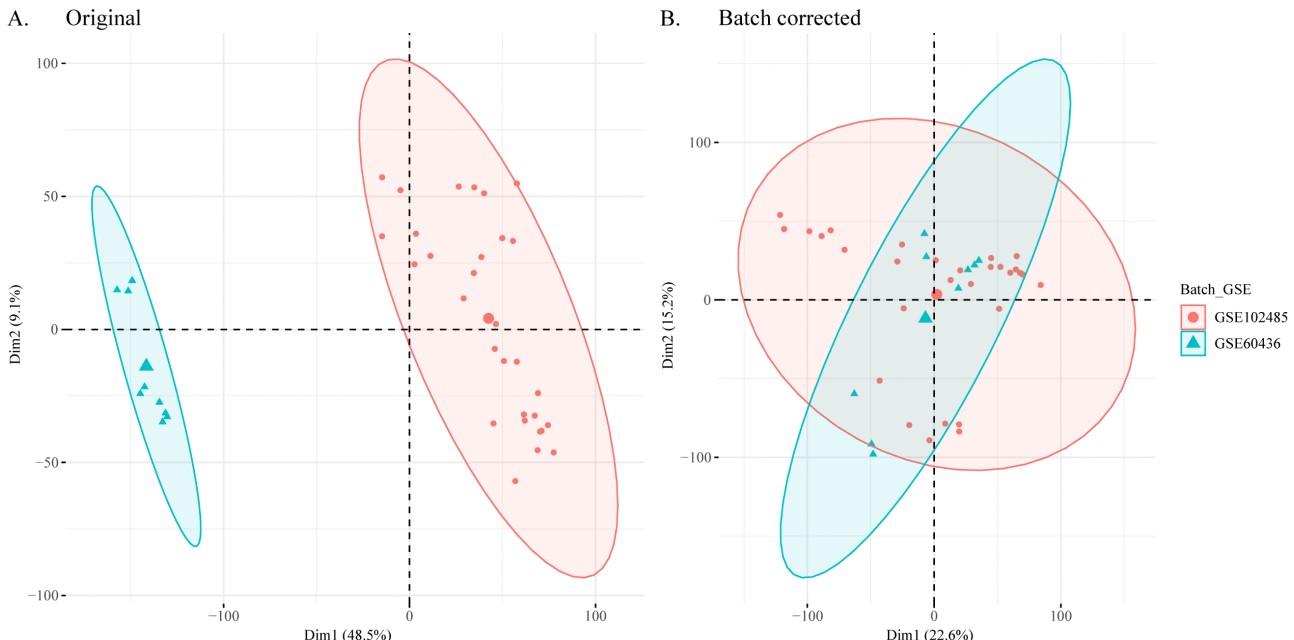

**Fig 1. Batch effect correction and data integration. (A)** Principal component analysis (PCA) plot showing the distribution of samples from GSE60436 and GSE102485 before batch effect removal. **(B)** PCA plot after batch effect removal using the sva package, demonstrating successful data integration.

was constructed based on 5 characteristic genes to predict the risk of patients with diabetic retina (Fig 4C). The results show that the prediction risk score has a significant predictive effect. The calibration curve shows that the prediction of the line chart model is accurate (Fig 4D). Moreover, the ROC curves of the 5 characteristic genes for predicting DR were analyzed, and the results showed that 8 characteristic genes were associated with favorable prognosis (Fig 4E–I).

### 3.4 Correlation and functional analysis of key diagnostic genes

Friends analysis showed that among all the essential genes, PLAU strongly interacted with the other genes (Fig 5A-B), while significant differences were observed in the expression of the 5 genes between the DR group and the normal group (Fig 5C). To further verify the expression level correlation and functional correlation of the 5 genes, we calculated the gene expression correlation of the 5 characteristic genes in all samples. The results showed that the highest correlation between PLAU and VGF gene expression among all samples was −0.89 (Fig 5D). At the PPI level, the most substantial protein interaction occurred between PLAU and PLAUR, and VGF interacted with other proteins through EGFR.

### 3.5 Differences in immune characteristics between DR and the normal retina

To evaluate the level of immune cell infiltration in DR, 22 states of immune cell infiltration were evaluated using CIBERSORT, and the Wilcoxon test was used to compare the abundance of different immune cell infiltrates. The results showed that the proportions of resting memory CD4 T cells and M2 macrophages were relatively high in DR (Fig 6A–B). These findings suggest that CD4 memory T cells and M2 macrophages are closely associated with DR pathogenesis.

### 3.6 Correlation analysis between key diagnostic genes and immune cell infiltration

Correlation analysis showed that there was a significant positive correlation between IL10RA and M2 macrophages ($r = 0.59$, $p < 0.001$) and between PLAU and monocytes ($r = 0.43$, $p < 0.001$). PLAUR was positively correlated with

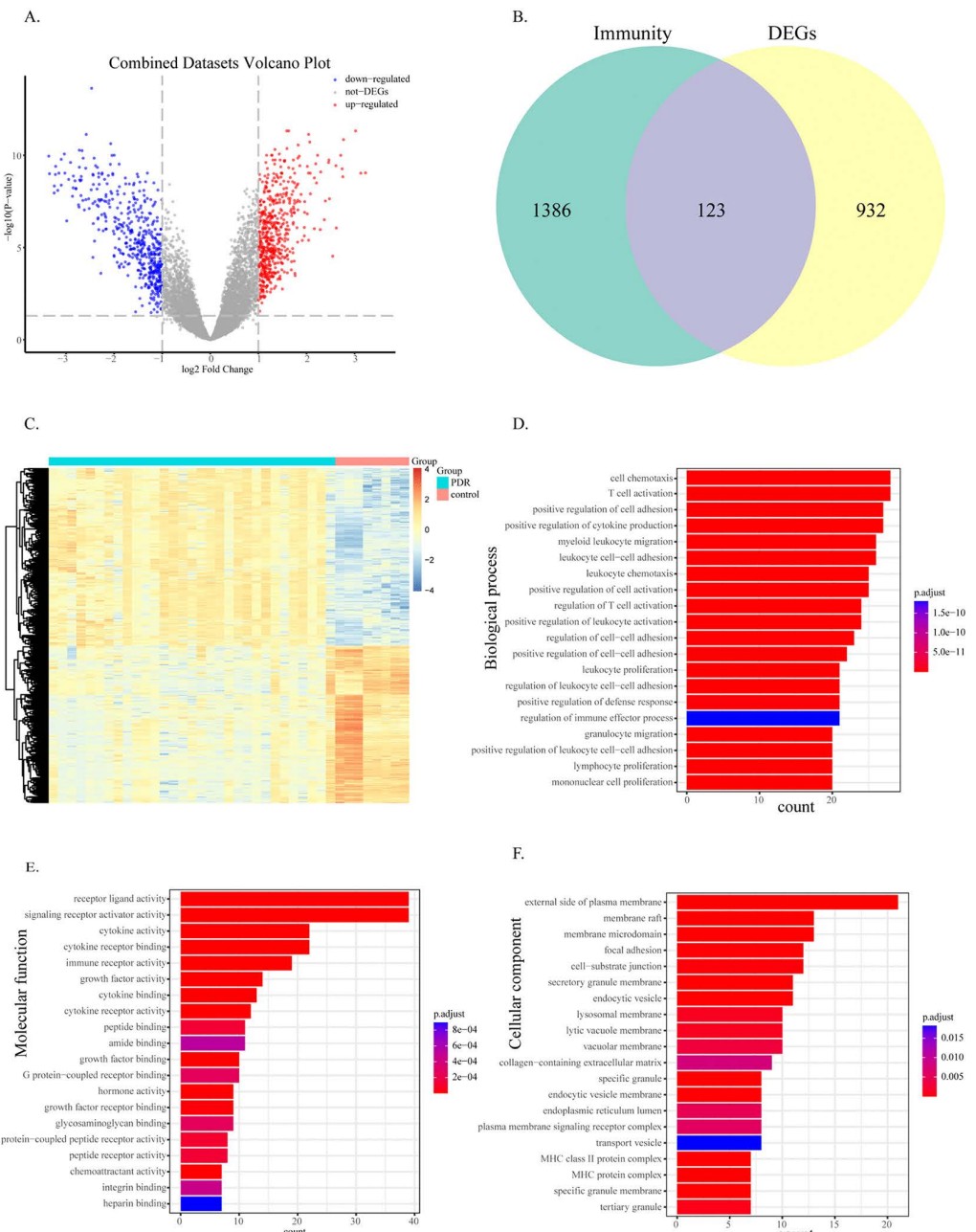

**Fig 2. Identification and functional enrichment of immune-related DEGs in DR. (A)** Volcano plot of differentially expressed genes (DEGs) between DR and normal retinal samples. Red dots: significantly upregulated genes (log2FC > 1, p < 0.05); blue dots: significantly downregulated genes (log2FC < −1, p < 0.05); gray dots: non-significant genes. **(B)** Venn diagram showing the intersection between all DEGs and immune-related genes, yielding 123 immune-related DEGs. **(C)** Heatmap of the top DEGs. Columns represent samples (blue: DR; pink: normal); rows represent genes (red: high expression; blue: low expression). **(D-F)** Gene Ontology (GO) enrichment analysis of immune-related DEGs, showing top enriched terms in biological processes **(D)**, cellular components **(E)**, and molecular functions **(F)**. Bar length indicates gene count, and color represents adjusted p-value.

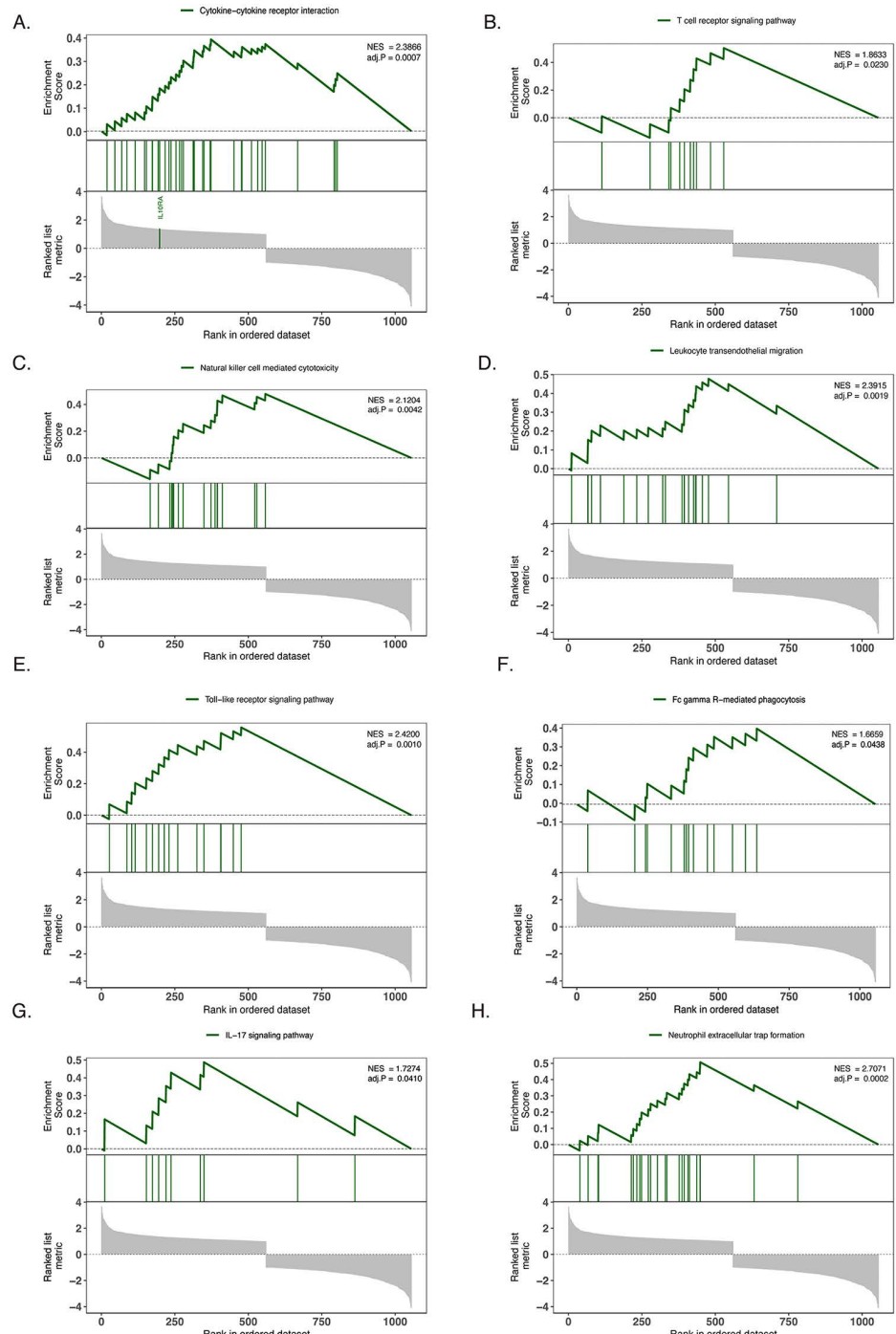

**Fig 3. Gene set enrichment analysis (GSEA) reveals immune pathway activation in DR. (A-H)** Enrichment plots showing significantly enriched pathways in DR samples compared to normal controls. Normalized enrichment score (NES) and p-value are shown for each pathway. Pathways include cytokine-cytokine receptor interaction **(A)**, T-cell receptor signaling **(B)**, NK cell-mediated cytotoxicity **(C)**, leukocyte transendothelial migration **(D)**, Toll-like receptor signaling **(E)**, Fc gamma R-mediated phagocytosis **(F)**, IL-17 signaling **(G)**, and neutrophil extracellular trap formation **(H)**.

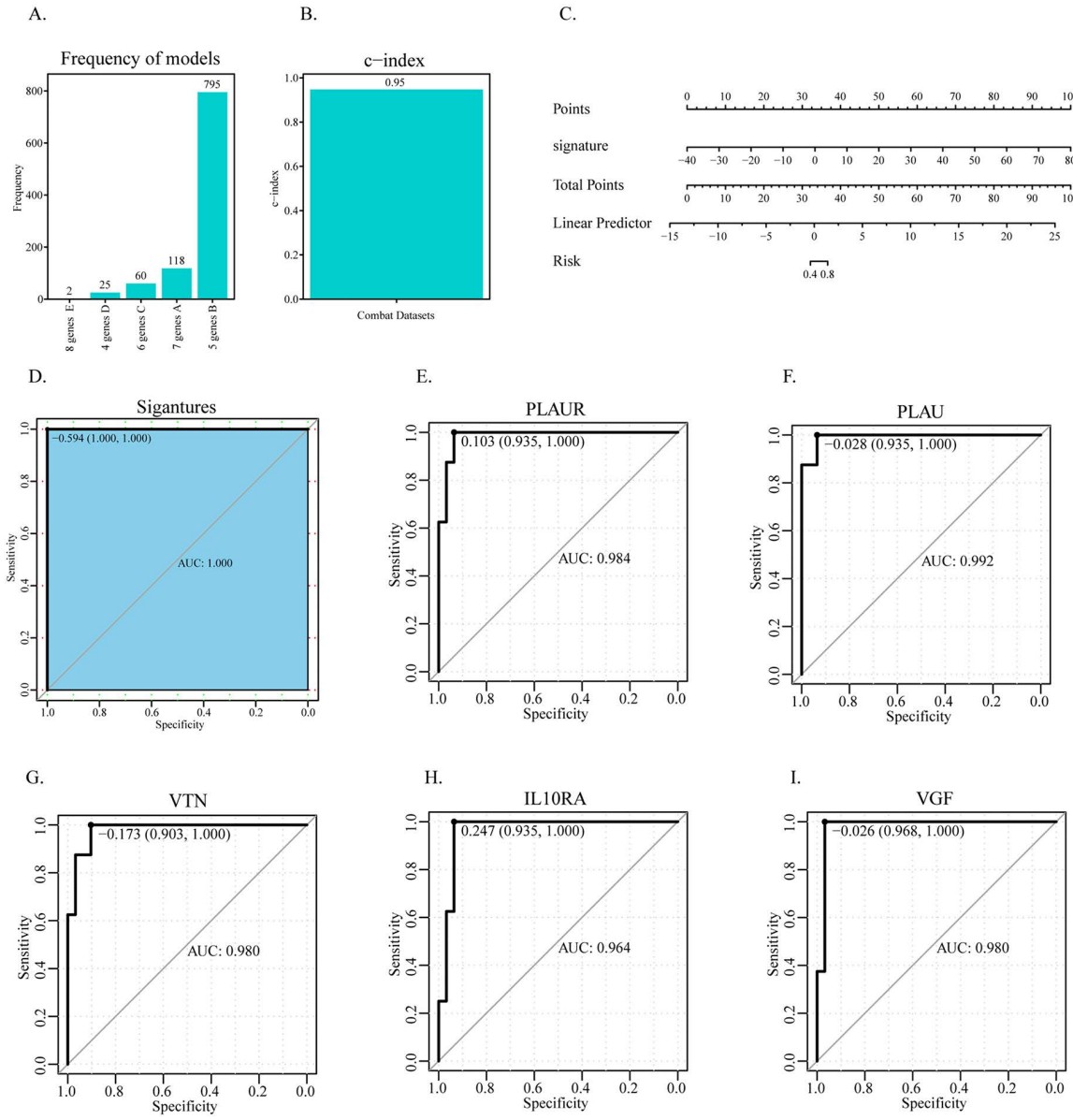

**Fig 4. Construction and validation of a diagnostic model for DR. (A)** LASSO regression coefficient profiles of immune-related genes. **(B)** Cross-validation plot for tuning parameter (λ) selection in the LASSO model. **(C)** Nomogram for predicting DR risk based on five key genes (PLAUR, PLAU, VTN, IL10RA, VGF). **(D)** Calibration curve assessing the predictive accuracy of the nomogram. The diagonal line represents ideal prediction; the pink line represents the nomogram performance. **(E-I)** Receiver operating characteristic (ROC) curves and area under the curve (AUC) values for each of the five key genes in discriminating DR from normal samples.

activated NK cells (r = −0.16, p < 0.001), initial CD4 + T cells (r = −0.28, p < 0.001) and plasma cells (r = −0.37). p < 0.001), all of which were negatively correlated. VGF was positively correlated with initial CD4 + T cells (r = 11, p < 0.001), whereas VTN was significantly positively correlated with plasma cells (r = 0.79, p < 0.001) (Fig 6 C). These findings collectively suggest that key diagnostic genes are closely linked to immune cell infiltration and may modulate immune responses in DR.

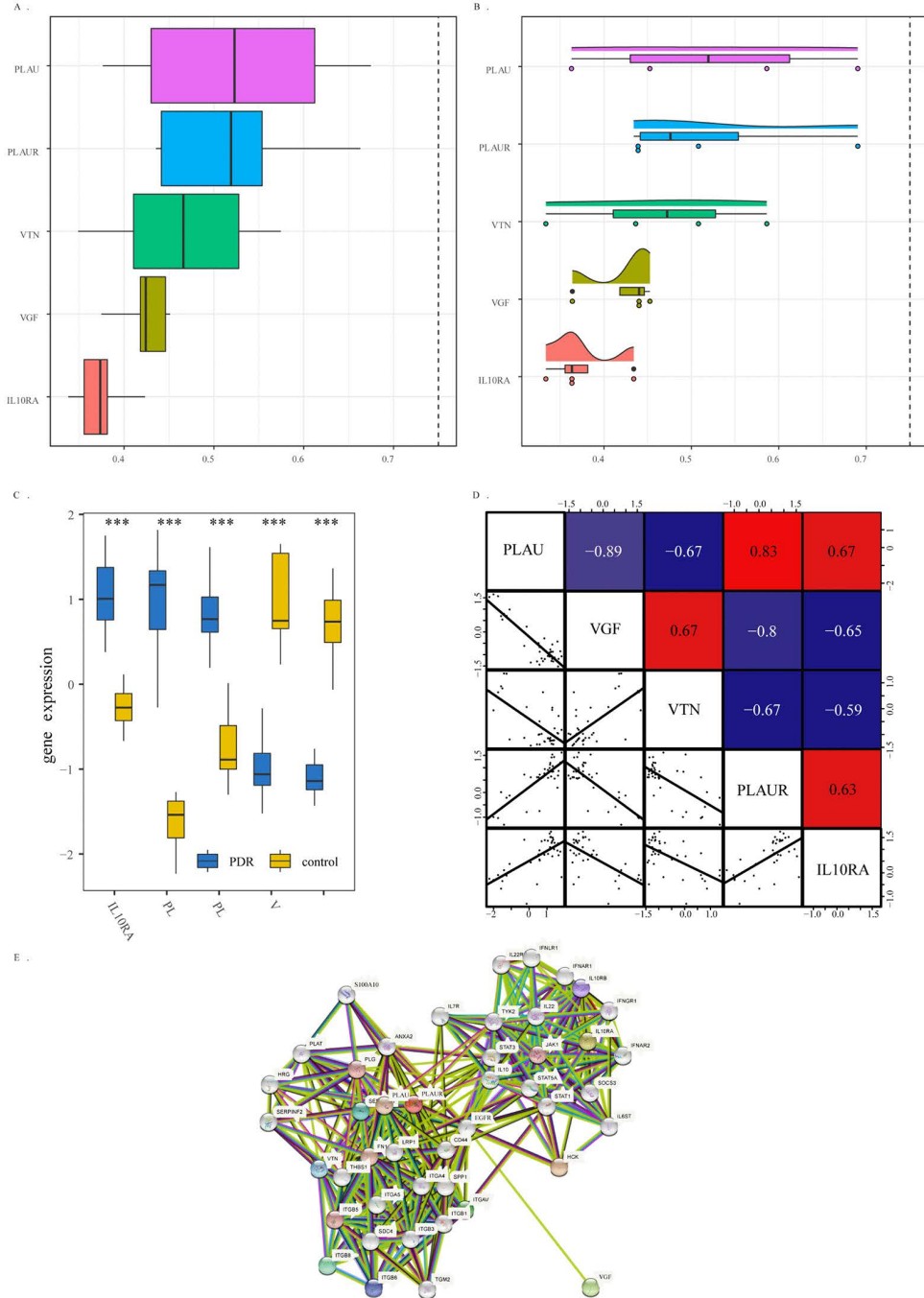

**Fig 5. Correlation, expression, and interaction analysis of key diagnostic genes. (A-B)** Friends analysis showing semantic similarity among the five key genes. **(C)** Expression levels of the five key genes in DR and normal retinal samples. *p<0.05, **p<0.01, ***p<0.001. **(D)** Correlation matrix of the five key genes across all samples. Numbers indicate Pearson correlation coefficients; blue represents positive correlation, red represents negative correlation. **(E)** Protein-protein interaction (PPI) network of genes associated with DR, constructed using the STRING database. Nodes represent proteins; edges represent interactions.

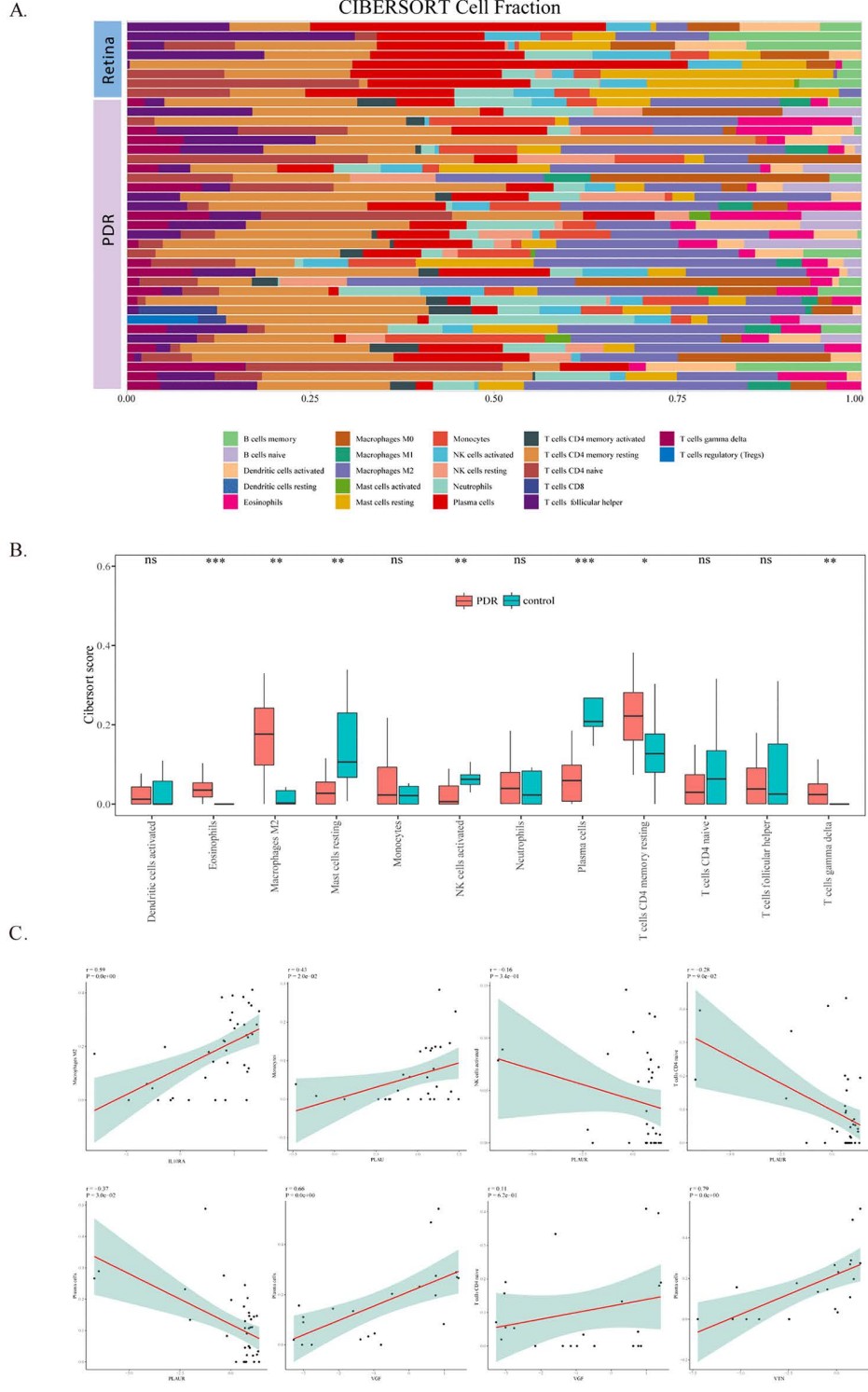

**Fig 6. Immune cell infiltration profiles in DR and correlation with key genes. (A)** Relative proportions of 22 immune cell subtypes in each DR and normal sample, estimated by CIBERSORT. **(B)** Comparison of immune cell infiltration abundance between DR and normal groups. Red: DR samples; blue: normal samples. *p < 0.05, **p < 0.01. **(C)** Correlation heatmap between the five key genes and infiltrating immune cell types. Red: positive correlation; blue: negative correlation. Color intensity reflects correlation strength.

### 3.7 Identification of two types of DR by immune-related DEGs

Using the R package ConsensusClusterPlus, two DR patterns (ClusterA and ClusterB) were identified using a consistent clustering method based on 5 characteristic genes (Fig 7A-B). ClusterA contained 6 samples, and ClusterB contained 25 samples.

### 3.8 Analysis of the difference between the two DR modes

The DEGs were analyzed in the DR model ClusterA and ClusterB (Fig 7C). The effects of DEGs between the two fatty acid metabolism patterns on the biological functions of patients were subsequently analyzed. GSEA showed that immune activation-related pathways, such as the B-cell receptor pathway, dendritic cell migration, and monocyte activation, were highly enriched in the ClusterB subgroup (Fig 7D). To analyze the differences between the two different modes of DR, the highly enriched biological signaling pathways in the two fatty acid metabolic subsets were explored using ssGSEA. The results revealed that G2M DNA replication checkpoints and other pathways were highly enriched in the ClusterA model, whereas STAT5 pathways were enriched in the ClusterB model (Fig 7E). These findings indicate that ClusterA and ClusterB exhibit distinct metabolic and immune signaling profiles, reflecting functional heterogeneity in DR.

### 3.9 Differences in immune characteristics between the two models

Compared with immune infiltration in the two subsets of DR, the infiltration of eosinophils, CD4+ effector memory cells, immature B cells, and Th2 cells was greater in the Cluster B subset (Fig 8A). Immune cell correlation analysis revealed that eosinophils were strongly correlated with effector B cells and memory B cells (Fig 8B).

### 3.10 Validation of key diagnostic genes

At 8 weeks post-STZ injection, blood glucose levels in the DM group remained significantly higher than those in the NC group (>300 mg/dL vs. ~100 mg/dL), confirming successful diabetes induction (Fig 9A). Conversely, body weight was significantly lower in the DM group throughout the experimental period (Fig 9B). Consistent with bioinformatics predictions, Western blot analysis revealed that protein levels of PLAU (53 kDa) and PLAUR (50 kDa) were significantly upregulated, while VGF (68 kDa) was significantly downregulated in the DM group compared to the NC group (Fig 9C and 9D).

## 4 Discussion

The incidence of diabetes is increasing due to modern lifestyle changes. In China, the prevalence increased from 10.9% to 12.4% between 2013 and 2018, with a 50.5% increase in diabetes and pre-diabetes combined [8,27]. This increase in diabetes is associated with various complications, among which DR is the most common, affecting up to 97% of patients with type 1 diabetes [28,29]. Current treatments for DR include laser therapy, intraocular injections, and vitrectomy, with anti-VEGF being commonly used [4,30,31]. However, caution is needed when anti-VEGF agents, especially bevacizumab, are used in severe patients with DR with pre-existing intraocular fibrosis, as they can lead to rapid vision loss [32]. Therefore, it is essential to more thoroughly investigate the targets and mechanisms involved.

In this study, the GEO datasets GSE60436 and GSE102485 were integrated to identify immune-related DEGs associated with DR (Fig 2). After batch effect correction and differential expression analysis, we identified 123 immune-related DEGs, further highlighting the critical role of immune-related pathways in DR (Fig 3). Functional enrichment analysis revealed significant enrichment of these genes in immune activation pathways, including T-cell receptor signaling, natural killer cell-mediated cytotoxicity, and IL-17 signaling pathways (Fig 4). Using LASSO regression analysis, we selected five key immune-related genes (PLAUR, PLAU, VTN, IL10RA, and VGF) and constructed a diagnostic model with strong predictive ability (Figures 5–6). Interestingly, our results indicated that all five genes were protective factors, and functional analysis revealed that among these five characteristic genes, PLAU was an essential functional gene (Figs 7-9).

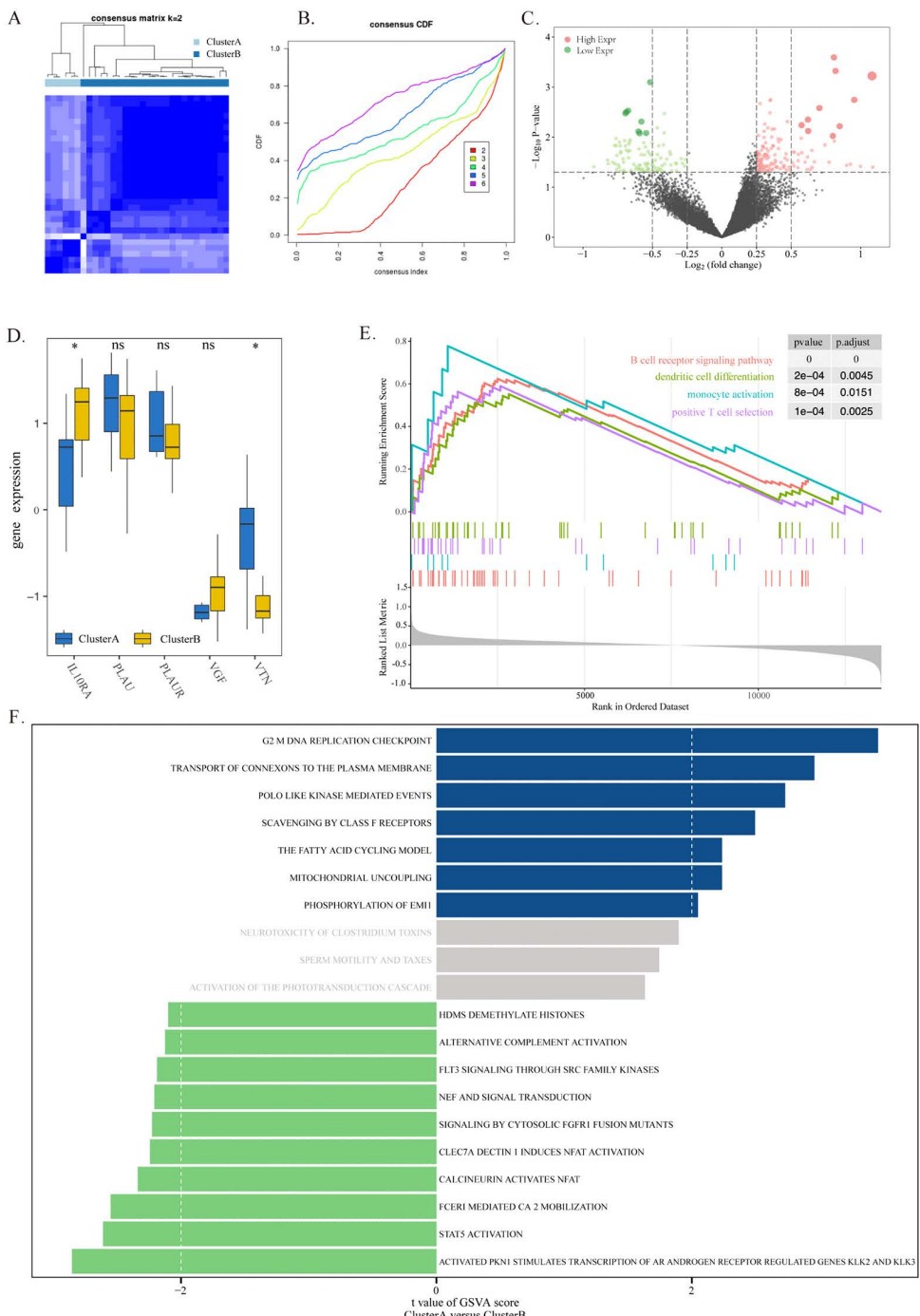

**Fig 7. Identification of two molecular subtypes of DR based on immune-related genes. (A)** Consensus clustering matrix for k = 2, showing clear separation of samples into two subtypes (Cluster A and Cluster **B**). **(B)** Consensus cumulative distribution function (CDF) plot for k = 2 to k = 9, indicating that k = 2 is the optimal choice. **(C)** Volcano plot of DEGs between Cluster A and Cluster **B.** Red: upregulated genes in Cluster B (log2FC > 1, p < 0.05); green: downregulated genes in Cluster B (log2FC < −1, p < 0.05); gray: non-significant genes. **(D)** Expression levels of the five key genes in Cluster A and Cluster **B**. Blue: Cluster A; yellow: Cluster **B**. *p < 0.05, **p < 0.01. **(E)** GSEA comparing Cluster A and Cluster B, showing enrichment of immune activation pathways in Cluster **B**. (F) ssGSEA scores for KEGG pathways in Cluster A and Cluster B, revealing distinct metabolic and immune signaling profiles between the two subtypes.

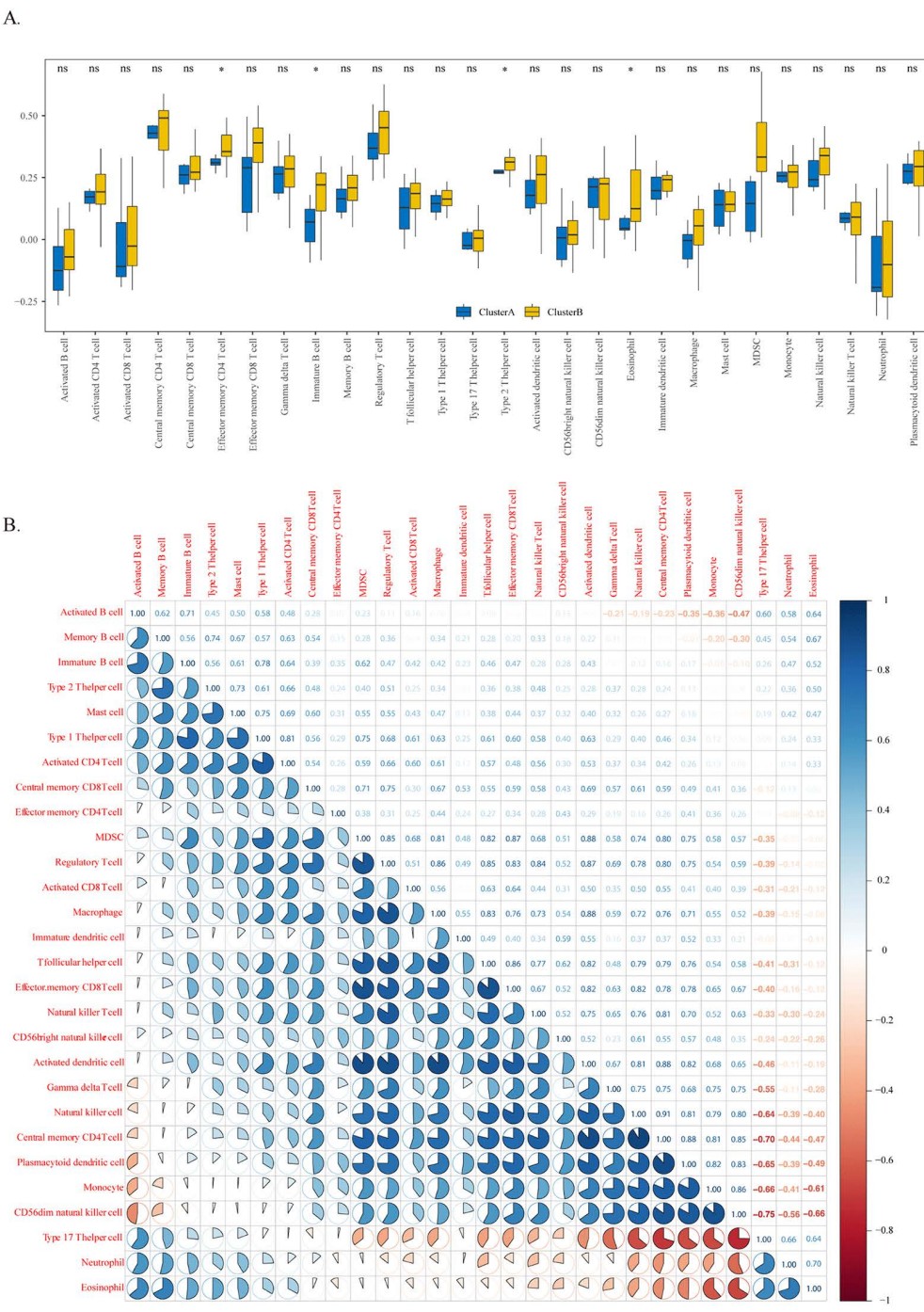

**Fig 8. Distinct immune infiltration patterns in DR molecular subtypes. (A)** Comparison of immune cell infiltration abundance between Cluster A and Cluster **B**. Blue: Cluster A; yellow: Cluster **B**. *p<0.05, **p<0.01. **(B)** Correlation matrix of immune cell types in all DR samples. Red: positive correlation; blue: negative correlation. Color intensity reflects correlation strength.

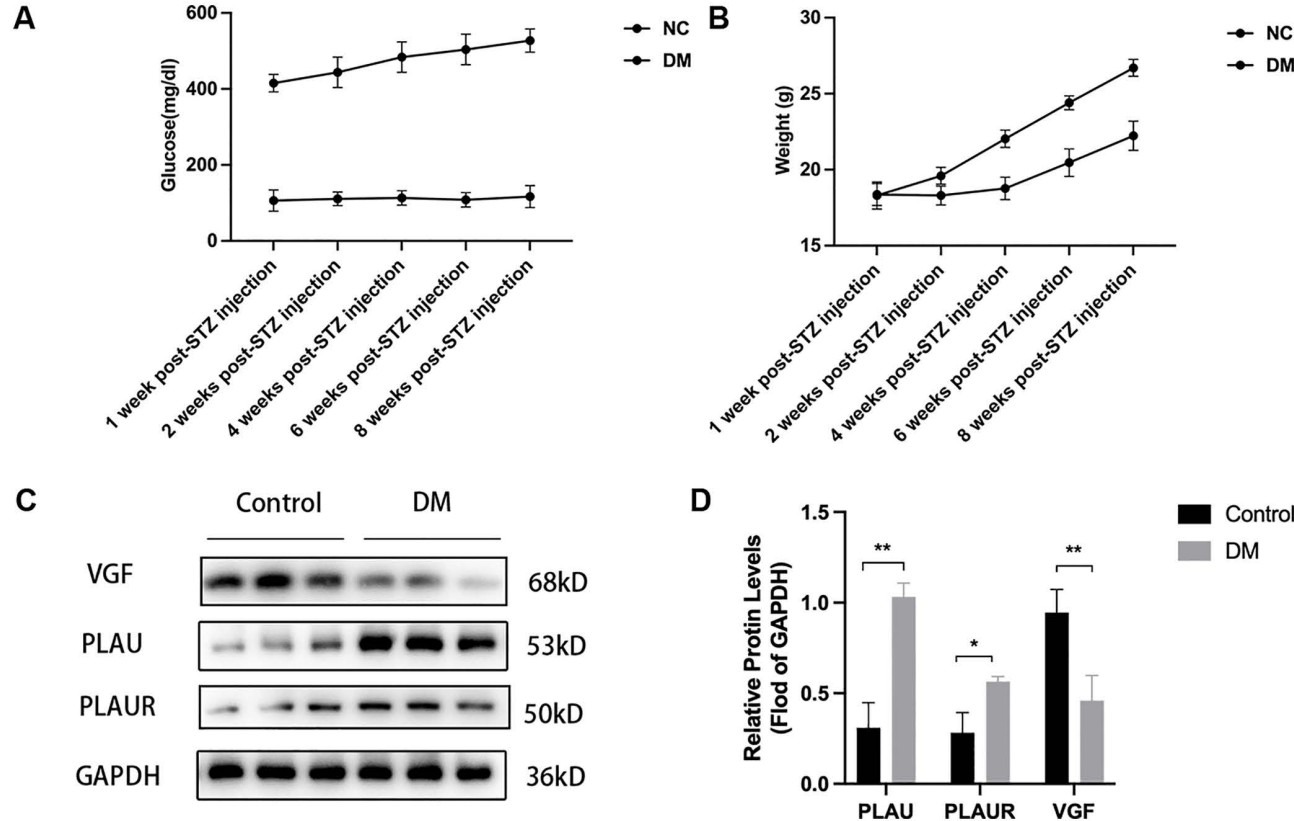

**Fig 9. Validation of key gene expression in an STZ-induced diabetic mouse model. (A)** Body weight changes in normal control (NC) and diabetic (DM) mice over 8 weeks post-STZ injection. **(B)** Representative Western blot images of PLAU, PLAUR, VGF, and GAPDH in retinal tissues from NC and DM mice. **(C)** Quantification of protein expression levels normalized to GAPDH. Data are presented as mean±SD. *p<0.05, **p<0.01 compared to NC group.

Urokinase-type plasminogen activator (PLAU) is a serine protease that converts plasminogen into plasmin [33]. Plasmin then hydrolyzes extracellular matrix proteins and activates growth factors [34]. Its receptor, PLAUR, participates in various physiological activities on the extracellular surface [35,36]. Although PLAU and PLAUR have been extensively studied in the context of cancer diagnosis and prognosis, their association with diabetic retinopathy (DR) remains underexplored [37,38]. Our analysis suggests that PLAU is a critical immune-related gene in DR and strongly correlated with the expression of the VGF gene. VGF, a transcript induced by NGF in PC12 cells, regulates energy, water-electrolyte balance, circadian rhythm, and reproductive activities [39]. Derivative peptides of VGF have been suggested to exert a protective effect on islet β-cells by modulating insulin biosynthesis and secretion [40,41]. Our analysis suggests that PLAU and VGF may also play an essential role in the occurrence and development of DR.

To further investigate the biological processes involved in differential genes, we conducted GSEA on the DEGs in the integrated GEO datasets. We found significant enrichment of immune-related pathways, including those involved in T-cell receptor signaling, natural killer cell-mediated cytotoxicity, and Toll-like receptor (TLR) signaling. T cells are essential for immune responses, with T-cell receptors (TCRs) playing a key role in activating immune processes [42]. Recently, T-cell-based immunotherapy has gained considerable attention in cancer research [43]. TLRs are intricately linked to T-cell responses by regulating both innate immunity and T-cell activation [44,45]. However, persistent TLR activation can lead to chronic inflammation or autoimmune responses, which are associated with diseases such as rheumatoid arthritis and

systemic lupus erythematosus [46,47]. Our findings suggest that the development and progression of diabetic retinopathy (DR) may be closely tied to the negative regulation of TLRs, which warrants further exploration [48,49].

In addition, we further examined immune cell infiltration in DR by performing PPI network analysis and using the CIBERSORT algorithm. Our findings revealed a significant increase in the infiltration of T cells and M2 macrophages in the retinas of patients with DR. T cells, which are central to immune responses, may contribute to DR pathology by modulating the retinal immune microenvironment [8,12]. M2 macrophages are involved in tissue repair and anti-inflammatory responses [50]. However, when excessively activated, they may contribute to chronic inflammation in DR [51]. These cells could also promote abnormal retinal vascular remodeling and structural changes, accelerating disease progression [52]. Additionally, immune cell infiltration patterns differed between the two molecular subtypes, Cluster A and Cluster B. Cluster A displayed heightened immune activity, particularly in T-cell and macrophage infiltration, which could be linked to more advanced retinal damage. In contrast, Cluster B exhibited lower levels of immune cell infiltration [53]. These results highlight the critical role of immune cells in the onset and progression of DR, with infiltration levels potentially correlated with clinical features and disease severity.

Although this study provides valuable insights into the immune-related mechanisms of DR, several limitations should be acknowledged. First, the analysis was based on publicly available datasets (GSE60436 and GSE102485), which may introduce selection bias due to the inherent characteristics of the original studies. Second, the clinical data primarily included patients with PDR, and the absence of NPDR samples limits the generalizability of our findings to early-stage DR. Third, the sample size of the GEO datasets is relatively small, which may affect the robustness of the identified molecular subtypes. Fourth, the animal experiments used a type 1 diabetes model (STZ-induced), whereas the bioinformatics analysis focused on type 2 diabetes-related datasets, potentially confounding the translational relevance. Finally, the functional roles of the hub genes were only validated at the protein level; further in vitro and in vivo experiments are needed to confirm their mechanistic involvement in DR pathogenesis. Future studies with larger, multi-center cohorts and more diverse stages of DR are warranted to validate and extend our findings.

## 5 Conclusion

In this study, immune-related genes and molecular subtypes involved in the pathogenesis of DR were explored using clinical data from two datasets. A LASSO regression revealed five key genes (IL10RA, PLAUR, PLAU, VTN, and VGF) as potential diagnostic biomarkers for DR. Immune activation-related pathways are closely associated with DR progression, particularly the infiltration of T cells and M2 macrophages.. Future research should further validate the roles of these genes in the retina and explore the specific functions of immune cells in DR.

## Supporting information

**S1 File. Original uncropped and unadjusted Western blot images.** The file contains all original Western blot images underlying the results presented in Figure 9. Each image is labeled with sample loading order, molecular weight markers, and the corresponding figure panel.
(PDF)

## Author contributions

**Conceptualization:** Xingtao Zhou.

**Data curation:** Yaojiani Cao.

**Formal analysis:** Lin Mu, Zhe Wang.

**Investigation:** Lin Mu.

**Software:** Zhe Wang.

**Validation:** Zhe Wang, Yaojiani Cao, Xuejun Wang.

**Writing – original draft:** Lin Mu.

**Writing – review & editing:** Xuejun Wang, Xingtao Zhou.

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
