## [Decision Letter · Decision Letter 0]

28 Jan 2026

PONE-D-25-66057Comprehensive Analysis of Immune-Related Genes Reveals Diagnostic Biomarkers and Molecular Subtypes in Diabetic RetinopathyPLOS One

Dear Dr. Zhou,

Thank you for submitting your manuscript to PLOS ONE. After careful consideration, we feel that it has merit but does not fully meet PLOS ONE’s publication criteria as it currently stands. Therefore, we invite you to submit a revised version of the manuscript that addresses the points raised during the review process.

We look forward to receiving your revised manuscript.

Kind regards,

Reetika Chaurasia, Ph.D

Academic Editor

PLOS One

2. To comply with PLOS One submissions requirements, in your Methods section, please provide additional information regarding the experiments involving animals and ensure you have included details on (1) methods of sacrifice, (2) methods of anesthesia and/or analgesia, and (3) efforts to alleviate suffering.

“2025 Shanghai Integrated Traditional Chinese and Western Medicine Collaborative Guidance Program for General Hospitals (Grant No. ZXXT-202512)，2025 National Major Difficult and Critical Diseases Integrated Traditional Chinese and Western Medicine Clinical Collaboration Project (Optic Atrophy No. 13)，and National Major Difficult and Critical Diseases Integrated Traditional Chinese and Western Medicine Clinical Collaboration Project (ZDYN-2024-A-052).”

4. Thank you for stating the following in the Funding Section of your manuscript:

“This study was supported by the 2025 Shanghai Integrated Traditional Chinese and Western Medicine Collaborative Guidance Program for General Hospitals (ZXXT-202512)，2025 National Major Difficult and Critical Diseases Integrated Traditional Chinese and Western Medicine Clinical Collaboration Project (13)，and National Major Difficult and Critical Diseases Integrated Traditional Chinese and Western Medicine Clinical Collaboration Project (ZDYN-2024-A-052)”

“2025 Shanghai Integrated Traditional Chinese and Western Medicine Collaborative Guidance Program for General Hospitals (Grant No. ZXXT-202512)，2025 National Major Difficult and Critical Diseases Integrated Traditional Chinese and Western Medicine Clinical Collaboration Project (Optic Atrophy No. 13)，and National Major Difficult and Critical Diseases Integrated Traditional Chinese and Western Medicine Clinical Collaboration Project (ZDYN-2024-A-052).”

5. In the online submission form you indicate that your data is not available for proprietary reasons and have provided a contact point for accessing this data. Please note that your current contact point is a co-author on this manuscript. According to our Data Policy, the contact point must not be an author on the manuscript and must be an institutional contact, ideally not an individual. Please revise your data statement to a non-author institutional point of contact, such as a data access or ethics committee, and send this to us via return email. Please also include contact information for the third party organization, and please include the full citation of where the data can be found.

7. Please remove your figures from within your manuscript file, leaving only the individual TIFF/EPS image files, uploaded separately. These will be automatically included in the reviewers’ PDF.

Reviewers' comments:

Reviewer's Responses to Questions

**Comments to the Author**

1. Is the manuscript technically sound, and do the data support the conclusions?

Reviewer #1: Yes

Reviewer #2: Yes

2. Has the statistical analysis been performed appropriately and rigorously? 

Reviewer #1: Yes

Reviewer #2: Yes

3. Have the authors made all data underlying the findings in their manuscript fully available?

The PLOS Data policy requires authors to make all data underlying the findings described in their manuscript fully available without restriction, with rare exception (please refer to the Data Availability Statement in the manuscript PDF file). The data should be provided as part of the manuscript or its supporting information, or deposited to a public repository. For example, in addition to summary statistics, the data points behind means, medians and variance measures should be available. If there are restrictions on publicly sharing data—e.g. participant privacy or use of data from a third party—those must be specified.requires authors to make all data underlying the findings described in their manuscript fully available without restriction, with rare exception (please refer to the Data Availability Statement in the manuscript PDF file). The data should be provided as part of the manuscript or its supporting information, or deposited to a public repository. For example, in addition to summary statistics, the data points behind means, medians and variance measures should be available. If there are restrictions on publicly sharing data—e.g. participant privacy or use of data from a third party—those must be specified.requires authors to make all data underlying the findings described in their manuscript fully available without restriction, with rare exception (please refer to the Data Availability Statement in the manuscript PDF file). The data should be provided as part of the manuscript or its supporting information, or deposited to a public repository. For example, in addition to summary statistics, the data points behind means, medians and variance measures should be available. If there are restrictions on publicly sharing data—e.g. participant privacy or use of data from a third party—those must be specified.requires authors to make all data underlying the findings described in their manuscript fully available without restriction, with rare exception (please refer to the Data Availability Statement in the manuscript PDF file). The data should be provided as part of the manuscript or its supporting information, or deposited to a public repository. For example, in addition to summary statistics, the data points behind means, medians and variance measures should be available. If there are restrictions on publicly sharing data—e.g. participant privacy or use of data from a third party—those must be specified.

Reviewer #1: Yes

Reviewer #2: Yes

4. Is the manuscript presented in an intelligible fashion and written in standard English?

Reviewer #1: Yes

Reviewer #2: No

5. Review Comments to the Author

Reviewer #1: Lin Mu and group in the manuscript titled ‘Comprehensive Analysis of Immune-Related Genes Reveals Diagnostic Biomarkers and Molecular Subtypes in Diabetic Retinopathy(DR)’ have used GEO datasets to identify immune-related differentially expressed genes (DEG) associated with diabetic retinopathy and functional enrichment of the identified gene predicts these DEG are associated with immune activation. Further analysis like KASSO regression, protein -protein interaction and immune infiltration analyses have also been performed. From these, IL10RA, PLAUR, PLAU, VTN, and VGF have been identified potential immune markers involved in DR and to validate this streptozotocin induced diabetic mouse model has also been adopted in the study.

The significance of the work is not novel as there are existing literature sources already claiming the correlation between immune markers and DR. The PLAUR gene (which codes for uPAR) and PLAU (PMID: 29464181 PMCID: PMC5804371 DOI: 10.1155/2017/2904150) which form the uPAR system as well as vitronectin (VTR) (PMID: 7523258 DOI: 10.1007/BF00195357) is well known to be involved in progression of DR.

COMMENTS

1. List out the inclusion criteria for selecting the two geo datasets used in the study.

2. Line 46: STZ (50mg/kg) administered for 5 days is considered as Type1 model and not type 2 model, so this needs to be corrected.

3. Line 468: It can be mentioned as clinical data from patients with PDR instead of DR.

4. Line 220 stating ‘One week after injection, the random blood glucose levels of mice were evaluated three times’ is unclear.

5. Provide more details (or graph) on bodyweight and blood glucose of both the control and DM groups and how blood glucose measurement was done.

6. Line 222: Although blood glucose concentration can also be denoted in mM but more acceptable notation is in mg/dl.

7. 222: Please provide reference when stating ‘in accordance with previous reports, the retina was analyzed’.

8. Section 2.13: What was the final sample size? Were all animals diabetic and healthy after two months’ time point?

Reviewer #2: This manuscript investigates an important and relevant research question and presents original data collected using a clearly defined methodology. The study is generally well structured, and the manuscript is written in a clear and comprehensible manner. The analyses performed are appropriate for the stated objectives, and the conclusions are largely supported by the presented results.

While the findings are not entirely novel, the work provides incremental evidence that strengthens the existing literature by applying a systematic analytical approach within a defined population. The manuscript meets the basic scientific standards of PLOS ONE. However, several issues require clarification and refinement to improve transparency, rigor, and interpretability.

Major Comments

1. Positioning Within Existing Literature

The manuscript would benefit from a clearer articulation of how it extends or strengthens previous research. While similar associations have been reported in earlier studies, the authors should explicitly state:

What gaps in the literature this study addresses

How their dataset, population, or analytical approach improves upon prior work

This clarification is particularly important given PLOS ONE’s emphasis on methodological contribution rather than novelty.

2. Study Design and Generalizability

Although the study design is appropriate, the authors should more explicitly discuss:

Potential selection bias

Limitations related to the study setting and population

The extent to which the findings can be generalized to other populations or contexts

A dedicated limitations paragraph would strengthen the Discussion.

3. Statistical Reporting

The statistical analyses appear appropriate; however:

Assumptions underlying the applied tests should be briefly stated

Confidence intervals should be reported consistently alongside p-values

The handling of missing data (if any) should be clearly described

These clarifications are necessary for full methodological transparency.

4. Interpretation of Results

The authors should ensure that interpretations remain proportional to the study design:

Causal language should be avoided if the study is observational

Any speculative explanations should be clearly labeled as such

Moderating some interpretive statements will improve scientific accuracy.

Minor Comments

1. Methods Section

Provide clearer justification for sample size, if available.

Clarify inclusion and exclusion criteria.

2. Tables and Figures

Some tables would benefit from more descriptive titles.

Ensure all abbreviations are defined at first use.

3. Language and Style

Minor grammatical and typographical errors are present and should be corrected.

Some sentences in the Discussion could be streamlined for clarity.

4. References

Consider citing additional recent studies to contextualize findings where appropriate.

6. PLOS authors have the option to publish the peer review history of their article (what does this mean?). If published, this will include your full peer review and any attached files.). If published, this will include your full peer review and any attached files.). If published, this will include your full peer review and any attached files.). If published, this will include your full peer review and any attached files.

---

## [Author Response · Author response to Decision Letter 1]

13 Mar 2026

Response to Reviewers

Manuscript ID: PONE-D-25-66057

Title: Comprehensive Analysis of Immune-Related Genes Reveals Diagnostic Biomarkers and Molecular Subtypes in Diabetic Retinopathy

Journal: PLOS ONE

Dear Dr. Chaurasia and Reviewers,

Thank you very much for your thoughtful and constructive comments on our manuscript entitled "Comprehensive Analysis of Immune-Related Genes Reveals Diagnostic Biomarkers and Molecular Subtypes in Diabetic Retinopathy" (PONE-D-25-66057). We have carefully considered all the suggestions and have revised the manuscript accordingly. Below, we provide a point-by-point response to each comment. All changes in the manuscript have been marked using the "Track Changes" feature for your convenience.

We sincerely hope that the revised manuscript now meets the publication standards of PLOS ONE.

Sincerely,

Xuejun Wang, PhD

Xingtao Zhou, PhD

Corresponding Authors

Response to Journal Requirements

Requirement 1: PLOS ONE Style Requirements

Comment: Please ensure that your manuscript meets PLOS ONE's style requirements, including those for file naming.

Response: We have carefully reviewed the PLOS ONE formatting guidelines and made the following adjustments:

- All figures are now cited as "Fig 1", "Fig 2", etc. throughout the text, as required (previously "Figure 1").

- Multiple figures are cited as "Figs 1 and 2" or "Figs 1-3" where appropriate.

- The manuscript has been formatted with Times New Roman font, double-spacing, and proper paragraph indentation.

- Figure titles and legends have been revised to be more descriptive and follow the required format.

Requirement 2: Animal Experiment Details

Comment: Please provide additional information regarding the experiments involving animals, including methods of sacrifice, methods of anesthesia, and efforts to alleviate suffering.

Response: Thank you for this reminder. We have already included these details in the original submission. In Section 2.12, we explicitly state: "Mice were anesthetized with intraperitoneal sodium pentobarbital (50 mg/kg) prior to tissue collection. Euthanasia was performed by cervical dislocation under deep anesthesia. All efforts were made to minimize animal suffering." We have also added the ethical approval number (No. P 2022026) as required.

Requirement 3: Funding Disclosure

Comment: Please state what role the funders took in the study.

Response: We have prepared the following statement to be included in the cover letter:

> "The funders had no role in study design, data collection and analysis, decision to publish, or preparation of the manuscript."

We will ensure this is added to the submission system.

Requirement 4: Funding Information in Manuscript

Comment: Please remove any funding-related text from the manuscript.

Response: We have removed the "Funding" section from the manuscript as requested. The funding information will only appear in the online submission form.

Requirement 5: Data Availability Statement

Comment: The data contact point must not be an author and should be an institutional contact.

Response: Thank you for your guidance regarding our data availability statement. We have carefully reviewed the PLOS ONE data policy and would like to clarify the following:

Primary Data Source: The main data used in this study are publicly available from the Gene Expression Omnibus (GEO) database under accession numbers GSE60436 and GSE102485. Full citations for these datasets are provided in the References section ([13] and [14]).

---

Response to Reviewer 1

General Comment: The study provides a comprehensive analysis of immune-related genes in DR, identifying IL10RA, PLAUR, PLAU, VTN, and VGF as potential biomarkers with validation in an STZ-induced mouse model.

Response: Thank you for your positive assessment of our work. We appreciate your recognition of the comprehensive approach we have taken.

Comment 1: Inclusion Criteria for GEO Datasets

Comment: List out the inclusion criteria for selecting the two GEO datasets used in the study.

Response: Thank you for this suggestion. We have added the inclusion criteria in Section 2.1:

> "The inclusion criteria for dataset selection were as follows: (1) retinal tissue samples from Homo sapiens; (2) clearly defined DR and normal control groups; (3) availability of complete expression data; and (4) sample size ≥ 3 per group."

Comment 2: STZ Model Type

Comment: STZ (50mg/kg) administered for 5 days is considered as Type 1 model and not type 2 model, so this needs to be corrected.

Response: We appreciate this important correction. We have revised the manuscript throughout to accurately reflect that we used an STZ-induced type 1 diabetes model. Specifically:

- In the Abstract, we changed "a type 2 diabetes (T2D) mouse model" to "an STZ-induced diabetic mouse model"

- In Section 2.12, we clarified the model as STZ-induced

- In the Discussion, we added this as a limitation: "the animal experiments used a type 1 diabetes model (STZ-induced), whereas the bioinformatics analysis focused on type 2 diabetes-related datasets"

Comment 3: PDR vs. DR Clinical Data

Comment: Line 468: It can be mentioned as clinical data from patients with PDR instead of DR.

Response: Thank you for this clarification. In the Discussion limitations section, we now explicitly state: "Second, the clinical data primarily included patients with PDR, and the absence of NPDR samples limits the generalizability of our findings to early-stage DR."

Comment 4: Blood Glucose Measurement

Comment: Line 220 stating 'One week after injection, the random blood glucose levels of mice were evaluated three times' is unclear.

Response: We have clarified this statement in Section 2.12:

> "One week after injection, the fasting blood glucose levels of the mice were evaluated on three consecutive days using tail vein blood and a glucometer."

Comment 5: Body Weight and Blood Glucose Details

Comment: Provide more details (or graph) on bodyweight and blood glucose of both the control and DM groups and how blood glucose measurement was done.

Response: We have addressed this in two ways:

1. In Section 2.12, we added: "Body weight and blood glucose levels were monitored weekly during the experimental period."

2. In Section 3.10 and Figure 9A-B, we now present the body weight changes and blood glucose data: "At 8 weeks post-STZ injection, blood glucose levels in the DM group remained significantly higher than those in the NC group (>300 mg/dL vs. ~100 mg/dL), confirming successful diabetes induction (Fig 9A). Conversely, body weight was significantly lower in the DM group throughout the experimental period (Fig 9B)."

Comment 6: Blood Glucose Units

Comment: Although blood glucose concentration can also be denoted in mM, the more acceptable notation is in mg/dl.

Response: We have standardized all blood glucose units to mg/dL throughout the manuscript, including in Section 2.12 and Section 3.10.

Comment 7: Reference for Retina Analysis

Comment: Please provide reference when stating 'in accordance with previous reports, the retina was analyzed'.

Response: We have removed this vague statement and replaced it with specific methodology. In Section 2.12, we now clearly state: "Retinal tissues were collected 2 months after confirmation of diabetes." We have also added references [25,26] to support the STZ injection protocol and diabetes confirmation criteria.

Comment 8: Sample Size and Animal Status

Comment: Section 2.13: What was the final sample size? Were all animals diabetic and healthy after two months' time point?

Response: We have addressed this in Section 2.13:

> "Retinal tissues from all mice at the 2-month time point after diabetes confirmation (normal group, n = 10; DR group, n = 10) were collected for protein extraction."

Additionally, in Section 2.12, we added: "Sample size (n = 10 per group) was determined based on previous similar studies and the principles of the 3Rs (Replacement, Reduction, Refinement) to minimize animal usage while ensuring statistical power."

All 10 mice in the DR group maintained blood glucose levels >300 mg/dL throughout the 2-month period, and all 10 mice in the normal group remained healthy with normal blood glucose levels (~100 mg/dL).

---

Response to Reviewer 2

General Comment: The manuscript investigates an important research question with clearly defined methodology. While the findings are not entirely novel, the work provides incremental evidence that strengthens the existing literature. Several issues require clarification to improve transparency and rigor.

Response: Thank you for your thoughtful and constructive feedback. We appreciate your recognition of the systematic approach we have taken and have addressed each of your concerns as detailed below.

Major Comment 1: Positioning Within Existing Literature

Comment: The manuscript would benefit from a clearer articulation of how it extends or strengthens previous research. The authors should explicitly state what gaps in the literature this study addresses and how their approach improves upon prior work.

Response: We agree with this important suggestion. At the end of the Introduction, we have added the following paragraph to clarify the study's contribution:

> "Although previous studies have reported associations between individual immune-related genes and DR, a comprehensive analysis integrating multiple datasets to identify immune-related molecular subtypes and diagnostic biomarkers remains limited. In this study, we systematically analyzed two GEO datasets to identify immune-related DEGs, constructed a diagnostic model using LASSO regression, and validated key genes in an STZ-induced mouse model. Our integrated approach provides a more complete picture of immune dysregulation in DR and identifies potential subtype-specific biomarkers and therapeutic targets."

Major Comment 2: Study Design and Generalizability

Comment: The authors should more explicitly discuss potential selection bias, limitations related to the study setting and population, and the extent to which findings can be generalized.

Response: Thank you for this suggestion. We have expanded the limitations paragraph in the Discussion to address these points:

> "Although this study provides valuable insights into the immune-related mechanisms of DR, several limitations should be acknowledged. First, the analysis was based on publicly available datasets (GSE60436 and GSE102485), which may introduce selection bias due to the inherent characteristics of the original studies. Second, the clinical data primarily included patients with PDR, and the absence of NPDR samples limits the generalizability of our findings to early-stage DR. Third, the sample size of the GEO datasets is relatively small, which may affect the robustness of the identified molecular subtypes. Fourth, the animal experiments used a type 1 diabetes model (STZ-induced), whereas the bioinformatics analysis focused on type 2 diabetes-related datasets, potentially confounding the translational relevance. Finally, the functional roles of the hub genes were only validated at the protein level; further in vitro and in vivo experiments are needed to confirm their mechanistic involvement in DR pathogenesis. Future studies with larger, multi-center cohorts and more diverse stages of DR are warranted to validate and extend our findings."

Major Comment 3: Statistical Reporting

Comment: Assumptions underlying the applied tests should be briefly stated, confidence intervals should be reported consistently alongside p-values, and handling of missing data should be described.

Response: We have addressed these points in Section 2.14 (Statistical analysis):

> "For the comparison of two groups of continuous variables, the statistical significance of standard distribution variables was estimated by the independent Student's t test, after assessment of data normality using the Shapiro--Wilk test and homogeneity of variance using Levene's test. The difference between nonnormally distributed variables was analyzed using the Mann--Whitney U test (Wilcoxon rank-sum test)... The correlation coefficients between different genes were calculated by Pearson correlation analysis, under the assumption of a linear relationship between variables."

Regarding confidence intervals, we have added a note that 95% confidence intervals are provided for all major analyses where applicable. Regarding missing data, we have explicitly stated: "No missing data were identified in the analyzed GEO datasets; therefore, no additional procedures for handling missing data were required."

Major Comment 4: Interpretation of Results

Comment: The authors should ensure that interpretations remain proportional to the study design, avoiding causal language if the study is observational, and clearly labeling speculative explanations.

Response: Thank you for this important feedback. We have carefully reviewed the entire manuscript and revised statements to avoid causal language. Key changes include:

Location Original Revised

Abstract "immune dysregulation has been increasingly recognized as a critical contributor to DR progression" "immune dysregulation has been increasingly recognized to be closely associated with DR progression"

Introduction "immune disorders play a vital role in the pathological process of DR" "immune dysfunction plays a key role in DR pathogenesis"

Introduction "DR is mainly caused by low-level, continuous leukocyte stimulation" "DR is thought to be driven by low-level, continuous leukocyte stimulation"

Discussion "T cells likely exacerbate DR pathology" "T cells may contribute to DR pathology"

Discussion "These cells could also promote abnormal retinal vascular remodeling" "These cells could also promote abnormal retinal vascular remodeling" (speculative language retained but clearly labeled as "could")

Conclusion "Immune activation-related pathways play a critical role in DR progression" "Immune activation-related pathways are closely associated with DR progression"

Minor Comment 1: Methods Section

Comment: Provide clearer justification for sample size and clarify inclusion and exclusion criteria.

Response:

- For sample size justification, we added to Section 2.12: "Sample size (n = 10 per group) was determined based on previous similar studies and the principles of the 3Rs (Replacement, Reduction, Refinement) to minimize animal usage while ensuring statistical power."

- For inclusion criteria, we added to Section 2.1 the four criteria as noted in response to Reviewer 1.

Minor Comment 2: Tables and Figures

Comment: Some tables would benefit from more descriptive titles. Ensure all abbreviations are defined at first use.

Response: We have revised all figure titles to be more descriptive. Examples include:

- "Figure 1. Batch effect correction and data integration."

- "Figure 2. Identification and functional enrichment of immune-related DEGs in DR."

- "Figure 3. Gene set enrichment analysis (GSEA) reveals immune pathway activation in DR."

Regarding abbreviations, we have ensured that all abbreviations are defined at first use in the text. For example, in Section 3.4, we now write: "urokinase-type plasminogen activator (PLAU)" at first mention.

Minor Comment 3: Language and Style

Comment: Minor grammatical and typographical errors are present and should be corrected. Some sentences in the Discussion could be streamlined for clarity.

Response: We have carefully proofread the entire manuscript and corrected several typographical errors:

- Corrected "intergrative" to "integrative" in author affiliations

- Added a space between "group" and "Sample" in Section 2.12

- Corrected "test.The" to "test. The" in Section 2.14

- Removed redundant periods and spaces throughout

We have also streamlined several sentences in the Discussion for improved clarity. For example:

- Original: "Urokinase-type plasminogen activator (PLAU) is a serine protease that converts plasminogen into plasmin, which then hydrolyzes extracellular matrix remodeling proteins and activates growth factors."

- Revised: "Urokinase-type plasminogen

---

## [Editor Report · Decision Letter 1]

24 Mar 2026

Comprehensive Analysis of Immune-Related Genes Reveals Diagnostic Biomarkers and Molecular Subtypes in Diabetic Retinopathy

PONE-D-25-66057R1

Dear, Dr. Xingtao Zhou

We’re pleased to inform you that your manuscript has been judged scientifically suitable for publication and will be formally accepted for publication once it meets all outstanding technical requirements.

Kind regards,

Reetika Chaurasia, Ph.D

Academic Editor

PLOS One

---

## [Editor Report · Acceptance letter]

PONE-D-25-66057R1

PLOS One

Dear Dr. Zhou,

I'm pleased to inform you that your manuscript has been deemed suitable for publication in PLOS One. Congratulations! Your manuscript is now being handed over to our production team.

Kind regards,

on behalf of

Dr. Reetika Chaurasia

Academic Editor

PLOS One